# Efficacy of Allogeneic and Xenogeneic Exosomes for the Treatment of Canine Atopic Dermatitis: A Pilot Study

**DOI:** 10.3390/ani14020282

**Published:** 2024-01-16

**Authors:** Sang-Won Kim, Kyung-Min Lim, Ssang-Goo Cho, Bokyeong Ryu, C-Yoon Kim, Seon Young Park, Kyungmin Jang, Jae Heon Jung, Cheolhyoung Park, Chulhee Choi, Jung-Hyun Kim

**Affiliations:** 1Department of Veterinary Internal Medicine, College of Veterinary Medicine, Konkuk University, Seoul 05029, Republic of Korea; everyday54@konkuk.ac.kr; 2Department of Stem Cell and Regenerative Biotechnology, Molecular & Cellular Reprogramming Center and Institute of Advanced Regenerative Science, Konkuk University, Seoul 05029, Republic of Korea; lmin0217@konkuk.ac.kr (K.-M.L.); ssangoo@konkuk.ac.kr (S.-G.C.); 3Department of Veterinary Physiology, College of Veterinary Medicine, Konkuk University, Seoul 05029, Republic of Korea; hobitmilk@jejunu.ac.kr (B.R.); vivavet@konkuk.ac.kr (C.-Y.K.); 4Department of Biomedical Informatics, College of Applied Life Sciences, Jeju National University, Jeju 63243, Republic of Korea; 5ILIAS Biologics Inc., Daejeon 34014, Republic of Korea; spark@iliasbio.com (S.Y.P.); jjkm604@naver.com (K.J.); jjung@iliasbio.com (J.H.J.); chpark@iliasbio.com (C.P.); cchoi@iliasbio.com (C.C.)

**Keywords:** canine atopic dermatitis, dog, exosome, immunomodulatory effect, mesenchymal stem cell, microbiome analysis

## Abstract

**Simple Summary:**

Canine atopic dermatitis is a multifactorial allergic skin disease that lacks permanent curative treatment, and a single treatment strategy does not show efficacy in all cases. Therefore, novel and innovative therapeutic options are urgently needed. Exosomes, one of the major types of extracellular vesicles, have been investigated as an alternative treatment for various diseases in humans; however, the efficacy or side effects of exosomes in the treatment of canine atopic dermatitis are elusive. In this study, we investigated the therapeutic potential of canine- and human-derived exosomes against canine atopic dermatitis using six experimental models randomly assigned to control, canine exosome, or human exosome groups. Our findings revealed that canine- and human-derived exosomes alleviated canine atopic dermatitis in clinical, immunological, and microbiological aspects and that the treatment with exosomes was well tolerated.

**Abstract:**

Canine atopic dermatitis (CAD) is a genetically predisposed inflammatory pruritic skin disease. The available treatments for CAD have several adverse effects and vary in efficacy, indicating the need for the development of improved treatments. In this study, we aimed to elucidate the therapeutic effects of allogeneic and xenogeneic exosomes on CAD. Six laboratory beagle dogs with CAD were randomly assigned to three treatment groups: control, canine exosome (cExos), or human exosome (hExos) groups. Dogs in the cExos and hExos groups were intravenously administered 1.5 mL of cExos (5 × 10^10^) and hExos (7.5 × 10^11^) solutions, respectively, while those in the control group were administered 1.5 mL of normal saline three times per week for 4 weeks. Skin lesion score and transepidermal water loss decreased in cExos and hExos groups compared with those in the control group. The exosome treatments decreased the serum levels of inflammatory cytokines (interferon-γ, interleukin-2, interleukin-4, interleukin-12, interleukin-13, and interleukin-31) but increased those of anti-inflammatory cytokines (interleukin-10 and transforming growth factor-β), indicating the immunomodulatory effect of exosomes. Skin microbiome analysis revealed that the exosome treatments alleviated skin bacterial dysbiosis. These results suggest that allogeneic and xenogeneic exosome therapy may alleviate CAD in dogs.

## 1. Introduction

Canine atopic dermatitis (CAD) is a genetically predisposed inflammatory skin disease with a prevalence rate of approximately 15% in the canine population [1,2,3,4]. CAD is characterized by pruritus and affects various body parts, among which the most affected regions include the face, ear pinnae, paws, perineum, ventrum, axillae, groin, and abdomen [5,6,7,8,9,10]. Primary skin lesions usually consist of erythematous macular or papular eruptions; however, most patients present with secondary lesions that occur due to self-induced trauma, including excoriation, self-induced alopecia, lichenification, and hyperpigmentation [5,8,10,11]. Moreover, the development of chronic lesions is often associated with secondary bacterial or yeast infections, such as *Malassezia* and *Staphylococcus* [7,12,13]. The clinical signs of CAD can be seasonal or nonseasonal, with or without seasonal exacerbation, depending on the allergens involved [7,8,10].

CAD poses a significant challenge and negatively affects the quality of life of affected dogs and their owners because of the lack of a permanent curative treatment and poor knowledge of its pathogenesis [4,14,15]. Pruritus is often the most common symptom in dogs; therefore, immediate and sustained control of pruritus is the primary therapeutic goal for managing CAD in veterinary practice [16,17]. However, the management of CAD is multimodal, and no single treatment strategy is likely to be successful in all cases [13,18]. Therefore, novel and innovative therapeutic options are urgently needed.

Mesenchymal stem cell (MSC) therapy has been extensively investigated as an alternative treatment for various diseases in humans; however, its safety and efficacy are controversial [19]. Additionally, the low viability of transplanted cells due to anoikis caused by the loss of cell adhesion is a primary limitation of MSC-based therapies [20,21,22]. However, recent studies suggest that the therapeutic effects of MSCs are attributable to the biological molecules they secrete in the form of microscopic particles called extracellular vesicles (EVs) [20].

Most eukaryotic and prokaryotic cells release EVs, which are important mediators of intercellular communication [23]. EVs are released by all cell types under physiological and pathological conditions and are present in all body fluids [24,25]. Exosomes are one of the major types of EVs, with diameters ranging from 40 to 100 nm [25,26,27], and they contain cell type-specific combinations of proteins, including enzymes, cytokines, and growth factors, as well as DNA, coding and noncoding RNAs, metabolites, and lipids [24,28]. Exosomes can travel through the bloodstream to cells in distant organs and easily fuse with recipient cells [20,29]. Additionally, exosomal contents are resistant to degradation via both endogenous and exogenous enzymes or RNases [29]. Exosomes do not require cell culture facilities and have easier storage and distribution than stem cells. These properties render them an attractive alternative to stem cells in human medicine. However, studies investigating the efficacy or side effects of exosomes derived from canines (cExos) or humans (hExos) in CAD treatment have not been reported.

Therefore, this study aimed to investigate the therapeutic potential of cExos and hExos in CAD models. We hypothesized that intravenous injection of allogeneic and xenogeneic exosomes would be tolerable and alleviate CAD. Additionally, we speculated that allogeneic exosomes would be more effective than xenogeneic exosomes in CAD treatment.

## 2. Materials and Methods

### 2.1. Study Subjects

Six laboratory beagle dogs with previously developed experimental CAD were used for the study [30]. The dogs had been sensitized epicutaneously twice a week for 12 weeks to house dust mite (HDM), *Dermatophagoides farinae* (*Der f*) (18M03, CITEQ biologics, Groningen, Netherlands, mixed with mineral oil). Successful sensitization was confirmed with *Der f*-specific IgE assay and intradermal test. After the end of the sensitization, six dogs were challenged twice a week for four weeks with the same HDM paste to induce clinical signs of atopic dermatitis. Immediately after the completion of the mite challenge, six CAD model dogs were included in this study. The animals were raised in individual kennels made of stainless steel in a room in the Konkuk Laboratory Animal Research Center. The kennels were cleaned once daily using high-pressure water, and the temperature and humidity of the room were maintained at 22–24 °C and 45–55%, respectively. During the experimental period, the animals were fed a conventional diet based on daily energy requirements and had ad libitum access to clean tap water. To avoid contamination, other dogs were not allowed to enter the room, and access was restricted to individuals directly involved in this study. Additionally, items such as bedding, clothes, toys, and bathing were prohibited from the animal house. The six CAD models were randomly assigned to allogeneic exosomes, xenogeneic exosomes, and control treatment groups. Every group had two dogs, and they were maintained throughout the entire study period. No concurrent medications were administered during the study period. This study was approved by the Institutional Animal Care and Use Committee (IACUC) of Konkuk University (IACUC approval no. KU21075, approval date 2 June 2021).

### 2.2. Production of cExos and hExos

Allogeneic exosomes (cExos) used in this study were isolated from canine adipose tissue-derived MSCs. The xenogeneic exosomes (hExos) were Expi293F cell-derived exosomes loaded with super-repressor IκB (srIκB). cExos and hExos were obtained following the methods reported in previous studies [31,32,33]. Briefly, both upstream and downstream processes were used in this study. In the upstream process, the cells were cultivated for 4 days using the WAVE method under blue-light exposure. Afterward, the conditioned media was subjected to ultrafiltration and purification to obtain the purified exosomes in the downstream process.

### 2.3. Counting and Immunoblotting cExos and hExos

Nanoparticle tracking analysis was performed to determine the size and particle concentration of EVs using an NS300 (Malvern Panalytical, Malvern, UK). Briefly, the samples were appropriately diluted (1:100–1:10,000) in particle-free phosphate buffer saline to achieve suitable concentrations. For immunoblotting, cells lysed in RIPA buffer or isolated exosomes were electrophorized using sodium dodecyl sulfate-polyacrylamide gel electrophoresis (SDS-PAGE) and transferred onto nitrocellulose membranes. Thereafter, the membranes were blocked with 5% skim milk in Tris-buffered saline containing 0.1% Tween-20 (TBS-T) and incubated with primary antibodies against target protein (srIκB, CRY2), exosome positive marker, and exosome negative marker overnight at 4 °C. The following primary antibodies were used in immunoblotting studies: srIκB, CRY2 (customized antibody from Abclon, Seoul, Republic of Korea), CD9, CD81 (SBI, Tokyo, Japan), TSG101, alix, GM130, calnexin (Abcam, Cambridge, UK), lamin B1, GAPDH (Santa Cruz Biotechnology, Dallas, TX, USA), and prohibitin (NOVUSBIO, Centennial, CO, USA). Following incubation with specific secondary antibodies, the blots were developed using Clarity and Clarity Max ECL western blotting substrates and visualized using a ChemiDoc imager.

### 2.4. Study Design

On day 0 (D0), skin lesion scoring was performed using a scale that was a modified form of the canine atopic dermatitis extent and severity index (CADESI-04), and transepidermal water loss (TEWL) was measured. Additionally, a skin swab was obtained from the same site for skin microbiome analysis. Blood samples were collected for circulating cytokine analysis.

Prior to the treatments, cExos and hExos were thawed and diluted with Dulbecco’s phosphate buffer saline. Thereafter, the animals were intravenously injected with 1.5 mL of cExos (5 × 10^10^ EVs) or hExos (7.5 × 10^11^ EVs) solution via the cephalic vein. The exosome dosage in this study was determined based on the effective dose and half-life of exosomes identified in our previous studies [32,34], which involved a different disease model. Dogs in the control group were intravenously administered 1.5 mL of normal saline (0.9% sodium chloride). The injection was repeated three times a week at two-, two-, and three-day intervals for four weeks. Overall, each experimental animal received a total of 12 doses during the 4-week period.

To assess the safety of cExos and hExos, the dogs were monitored for signs of allergic reaction or anaphylaxis for 30 min following every injection, and physical examinations were performed daily during the study period. Lesion scoring was performed after four and eight doses of the treatments and at the end of the experiment using the modified CADESI-04 scale [35]. Additionally, at baseline and the end of the study period (on day 28; D28), TEWL measurement, skin microbiome analysis, and circulating cytokine analysis were performed (Figure 1).

### 2.5. Evaluation of Skin Lesions

The severity of skin lesions was scored on days 0, 10, 19, and 28 using a modified CADESI-04 scale [35] by a veterinarian who was blinded to the study details. Erythema, lichenification, and excoriation lesions of the right groin were scored from 0 to 3, with a maximum score of 9.

### 2.6. Evaluation of Skin Barrier Function

The water barrier integrity of the stratum corneum was evaluated based on the TEWL measured in the right groin region on D0 and D28 using a VapoMeter^®^ (Delfin Technologies, Kuopio, Finland), according to the manufacturer’s instructions [36]. All measurements were performed in the same room in which the dogs were raised. Mean values were calculated after obtaining five consecutive values.

### 2.7. Serum Cytokine Analysis

Blood samples (5 mL) were obtained from the jugular vein, followed by centrifugation at 3500 rpm for 15 min to collect serum. Serum levels of inflammatory cytokines were measured on D0 and D28 using commercially available canine ELISA kits for interferon (IFN)-γ (DY781B, R&D Systems, Minneapolis, MN, USA), interleukin (IL)-2 (DY1815, R&D Systems), IL-4 (DY754, R&D Systems), IL-10 (DY735, R&D Systems), IL-12 (DY1969, R&D Systems), IL-13 (SEA060Ca, Cloud-Clone Corp., Katy, TX, USA), IL-31 (ECI0041, ABclonal, Woburn, MA, USA), and TGF-β (DB100B, R&D Systems), according to the manufacturer’s instructions.

### 2.8. Skin Microbiome Analysis

Skin swab samples were collected from the right groin before and after the treatment (D0 and D28). The swab samples were stored at −80 ℃ until bacterial DNA extraction, PCR amplification, and 16S rRNA sequencing. PCR amplification of the extracted DNA was performed using fusion primers targeting the V3–V4 regions of the 16S rRNA gene. Sequencing was performed at CJ Bioscience, Inc. (Seoul, Republic of Korea) using MiSeq (Illumina, San Diego, CA, USA), an integrated next-generation sequencing instrument. The primer sequences used for the PCR and data analysis after sequencing were the same as those in a previous study [30].

### 2.9. Statistical Analysis

The sample size was too small to perform statistical analysis, and only a descriptive assessment was performed. For comparison between groups, the change in baseline was calculated because the pretreatment values were different in all dogs. The change from baseline value was defined as the difference between post-treatment and pretreatment values. All values are expressed as the mean ± standard error.

## 3. Results

### 3.1. Evaluation of Skin Lesions

Treatment with cExos (allogeneic exosomes) and hExos (xenogeneic exosomes) remarkably ameliorated skin lesions in the right groin, including erythematous edema, excoriation, and lichenification, compared with that in the control group (Figure 2). Additionally, cExos and hExos treatments caused a gradual decrease in the modified CADESI-04 score. Specifically, the change in modified CADESI-04 score (post-treatment score—baseline score) was −0.5 ± 0.5, −4.0 ± 2.0, and −3.5 ± 0.5 in the control, allogeneic exosome, and xenogeneic exosome groups, respectively (Figure 3).

### 3.2. Evaluation of Skin Barrier Function

TEWL was measured in each group before treatment (D0) and after all treatment doses (D28). The TEWL score was reduced in the cExos and hExos groups compared to that in the control group, indicating that cExos and hExos treatment improved skin barrier integrity in the dogs. Specifically, the change in TEWL value (post-treatment value—baseline value) was 1.9 ± 2.5, −34.65 ± 1.25, and −20.0 ± 13.0 g/m^2^/h in the control, cExos, and hExos groups, respectively (Figure 4).

### 3.3. Serum Cytokine Levels

Next, we evaluated the changes in the serum levels of IFNγ, IL-2, IL-4, IL-10, IL-12, IL-13, IL-31, and TGFβ (Figure 5; Table 1). Treatment with the exosomes decreased the levels of inflammatory cytokines and increased the serum levels of anti-inflammatory cytokines compared to that in the control group.

### 3.4. Skin Microbiome Analysis

A total of 420,237 and 658,452 reads were obtained from the skin swab samples of all the subjects before and after treatment, respectively. Compared with before treatment, the number of operational taxonomic units (OTUs) increased in cExos and hExos groups after treatment (Table 2). The numbers of normalized reads were 61,927 and 30,956 in all groups before and after the treatments, respectively. The alpha diversity indices were calculated after normalizing the reads before and after the treatment in each sample (Table 2). Changes in OTUs and Chao1 and Shannon indices after the treatments were calculated (Table 3) and are presented in Figure 6. Also, principal coordinate analysis (PCoA) plots from the generalized UniFrac metric were constructed (Figure 7).

## 4. Discussion

Currently, available treatment strategies for CAD include symptomatic treatments and specific interventional therapies [37,38]. Symptomatic therapy includes glucocorticoids; calcineurin inhibitors such as cyclosporine and tacrolimus; the Janus kinase inhibitor oclacitinib; and the monoclonal interleukin-31 antibody, lokivetmab [18,39]. Glucocorticoids are inexpensive and show rapid action; however, they can cause side effects, such as polyuria/polydipsia, polyphagia, muscle atrophy, skin atrophy, and iatrogenic hyperadrenocorticism [10,38,40]. Additionally, the side effects of cyclosporine were reported in 55% of the 15 clinical trials of CAD, especially gastrointestinal side effects such as vomiting and diarrhea [41,42]. Although long-term administration of oclacitinib seems to be well tolerated, new or exacerbated existing neoplasia has been observed in dogs treated with oclacitinib [43,44]. Lokivetmab showed more pronounced antipruritic efficacy than cyclosporine but had a lower anti-inflammatory effect [45,46]. Interventional therapies include allergen-specific immunotherapy, which is the only etiological treatment for CAD [47]. Various routes have been used to administer allergen-specific immunotherapy; however, the risk of anaphylaxis must be considered [48,49].

Treatment with cExos and hExos ameliorated skin lesions in dogs with CAD compared with that in the control group, as evidenced by a decrease in modified CADESI-04 scores in the cExos and hExos groups. Additionally, cExos and hExos treatments reduced TEWL, indicating that exosome therapy improved skin barrier function. Moreover, exosome treatments decreased the serum levels of inflammatory cytokines, including IFNγ, IL-2, IL-4, IL-12, IL-13, and IL-31, and increased the levels of anti-inflammatory cytokines, including IL-10 and TGFβ. This trend was more pronounced in the exosome treatment groups than in the control group, indicating that exosome therapy has an immunomodulatory effect. Given that IL-31 is a potent pruritogenic cytokine, the decrease in IL-31 levels emphasizes the antipruritic effect of exosomes [50].

The alpha diversity indices represent the species richness or evenness of the microbiome. The findings of this study showed an increase in OTUs and Chao1 and Shannon indices after exosome treatment. Consistent with our findings, a previous study has shown that allergic dogs have lower species diversity in the skin microbiome than healthy dogs [51]. Therefore, the results of this study suggest that the exosome treatments improved skin microbiome dysbiosis in the CAD model.

MSC-derived EVs exhibit the same functions as stem cells, including anti-inflammatory and proregenerative activities [52]. Additionally, MSC-derived EVs are smaller and less immunogenic than MSCs, making them safer alternatives to cell injections [52]. In human medicine, the therapeutic effects of MSCs have been largely attributed to the EVs they secrete, supporting the regenerative and immunomodulatory capabilities of MSCs [53]. However, this approach is still in the early stages of research, and there has only been one study on the therapeutic use of MSC-derived exosomes in veterinary medicine [24]. Canine MSC-derived microvesicles or exosomes significantly promote the healing of cutaneous wounds, collagen synthesis, and vascularization at wound sites in dogs [54]. Although the therapeutic effects of autogenous or allogeneic adipose-derived MSCs on CAD [55,56] or experimental CAD models [57] have been examined, studies have yet to examine the effects of exosome therapy on CAD. Additionally, the therapeutic effect of xenogeneic exosomes in dogs with CAD is yet to be reported.

In this study, we investigated the therapeutic effects of human-derived exosomes isolated from the Expi293F cell line, which was derived from the HEK293 cell line. These are human embryonic kidney cell lines that are easy to manipulate and show high growth capacity; therefore, they are commonly used to produce therapeutic proteins or exosomes in human medicine [33,58]. We used the engineered exosome loaded with srIκB, a nondegradable form of IκB that prevents the nuclear translocation of NFκB [59]. NFκB is one of the most important regulators of proinflammatory gene expression and is highly activated at sites of inflammation in diverse diseases [60]. The effects of srIκB loaded exosomes for sepsis, acute kidney injury, and preterm birth have been confirmed in several studies [32,61,62].

Nevertheless, the immunogenicity of xenogeneic transplantation is a serious issue that should be considered in therapeutic research. The intravenous administration of Expi293F-derived EVs to mice did not induce significant toxicity [63]. Similarly, cExos and hExos treatments (allogeneic and xenogeneic exosomes) were well tolerated in the present study, as evidenced by the absence of any allergic reaction, anaphylaxis, and abnormal findings. However, the duration of this study was only 4 weeks; therefore, further studies are necessary to verify the long-term safety of xenogeneic exosomes.

In the present study, it was difficult to compare the therapeutic efficacy of canine- and human-derived exosomes due to differences in their administration dosage, the origin of cells, and the small number of study subjects. A comparative study of the regenerative effects of allogeneic and xenogeneic adipose tissue-derived EVs on full-thickness skin wounds in rats showed that both exosomes had almost the same effects [64]. Collectively, these results indicate that EVs from the same tissue carry similar contents and that the species might not be the main factor for EV function. However, further studies are necessary to verify the efficacy of allogeneic and xenogeneic exosomes in CAD.

The xenogeneic exosomes used in this study were not derived from human MSC but from an embryonic cell line that was previously stored. However, the xenogeneic exosomes had a similar immunomodulatory effect as the allogeneic MSC-derived exosomes in CAD. Cell line-derived exosomes are widely used to produce human industrial bioproducts due to the ease of large-scale production and extraction from existing cells [58]. The large-scale production of therapeutic exosomes is an unexplored field in veterinary medicine, indicating the need for further studies.

Although promising results were obtained, this study has some limitations. For example, statistical analysis and comparisons could not be performed due to the small number of study subjects; therefore, a definite conclusion on the efficacy of exosomes for CAD cannot be inferred. However, the positive findings of this study support the need for larger placebo-controlled trials involving patients with CAD. Additionally, we did not compare the histopathology and immunohistochemistry at the lesion sites before and after treatment in this study to minimize invasive tests. However, we evaluated changes in TEWL, circulating cytokines, and the skin microbiome. Furthermore, an experimental CAD model was used in this study; therefore, the effects of exosomes on patients with spontaneous CAD cannot be predicted. Further studies are required to verify the efficacy of exosomes in patients with CAD.

## 5. Conclusions

Conclusively, the findings of the study showed that allogeneic and xenogeneic exosome treatment was feasible and well tolerated in the experimental CAD model, with no evidence of adverse events. Additionally, this study provides a basis for future large-scale studies to confirm the long-term safety and efficacy of allogeneic and xenogeneic exosomes in CAD.

## Figures and Tables

**Figure 1 animals-14-00282-f001:**
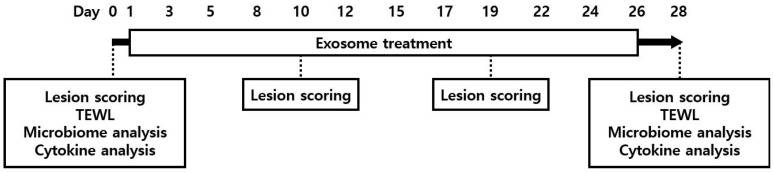
An outline of this study. Study subjects were treated with 12 doses of canine exosomes (cExos) and human exosomes (hExos) on days 1, 3, 5, 8, 10, 12, 15, 17, 19, 22, 24, and 26. Animals were evaluated on day 0 (D0) for baseline values and on D28 after treatment. During this study, lesion scoring was performed after four and eight doses of treatment (before the injection of D10 and D19) using the modified CADESI-04 scale. CADESI, canine atopic dermatitis extent and severity index; TEWL, transepidermal water loss.

**Figure 2 animals-14-00282-f002:**
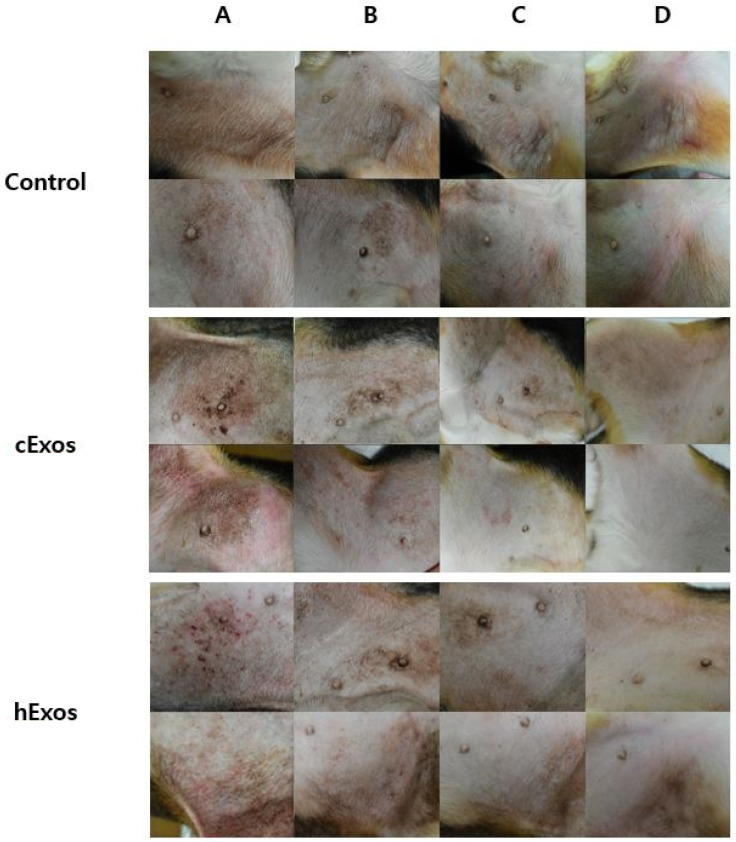
Representative skin lesions in the right groin area of dogs in control, canine exosome (cExos), and human exosome (hExos) groups. Images show the time course of skin lesions on D0 (**A**), D10 (**B**), D19 (**C**), and D28 (**D**).

**Figure 3 animals-14-00282-f003:**
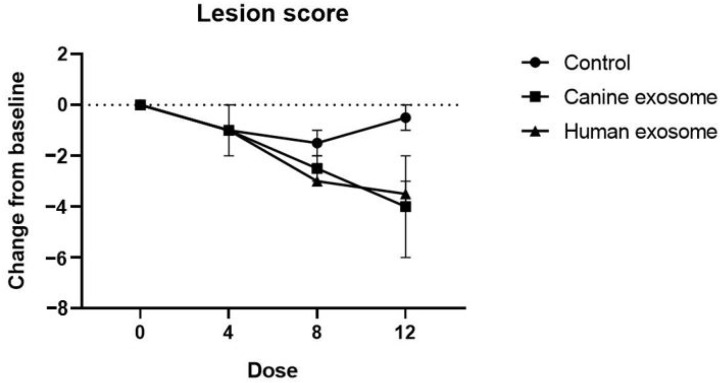
Changes in lesion scores at different time points compared with the baseline values. Dose 0, D0; Dose 4, D10; Dose 8, D19; Dose 12, D28. All values are expressed as the mean ± standard error.

**Figure 4 animals-14-00282-f004:**
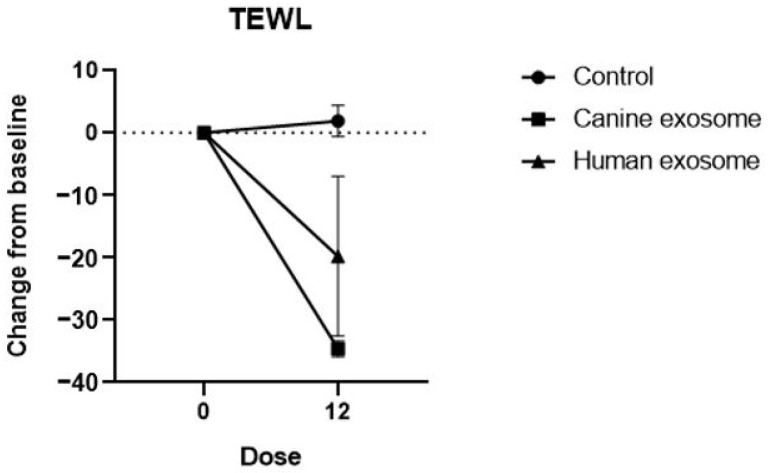
Changes in TEWL values at specific time points compared with the baseline values. Dose 0, D0; Dose 12, D28. All values are expressed as the mean ± standard error. TEWL, transepidermal water loss.

**Figure 5 animals-14-00282-f005:**
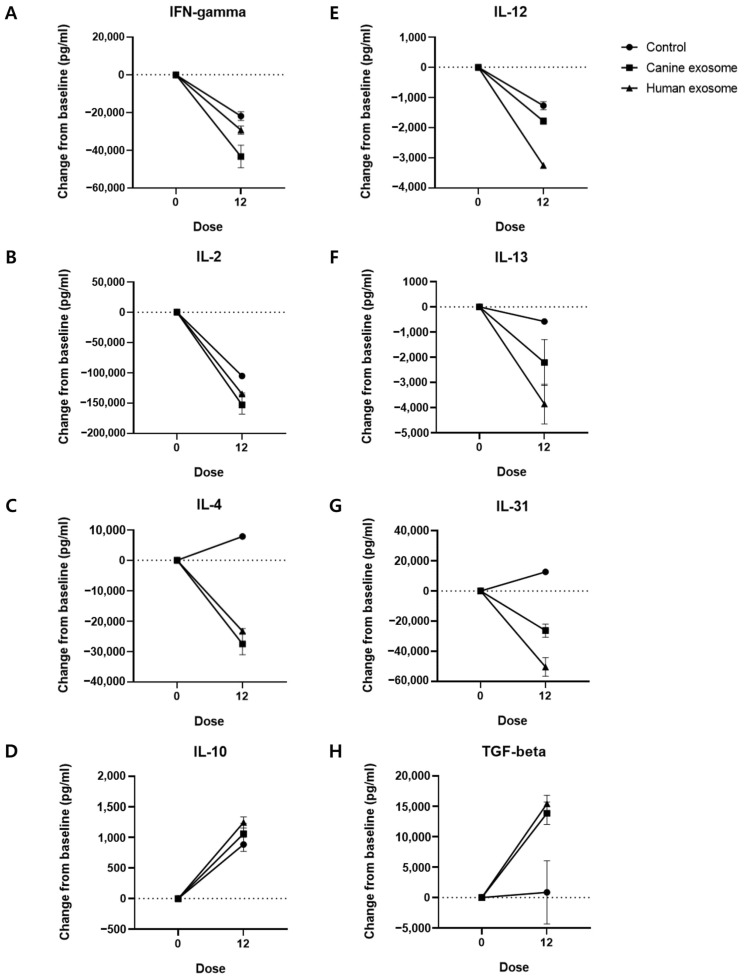
Serum cytokine analysis. Changes in serum levels of (**A**) IFN-γ, (**B**) IL-2, (**C**) IL-4, (**D**) IL-10, (**E**) IL-12, (**F**) IL-13, (**G**) IL-31, and (**H**) TGF-β compared with the baseline values. Dose 0, D0; Dose 12, D28. All values are expressed as the mean ± standard error. IFN, interferon; IL, interleukin; TGF, transforming growth factor.

**Figure 6 animals-14-00282-f006:**
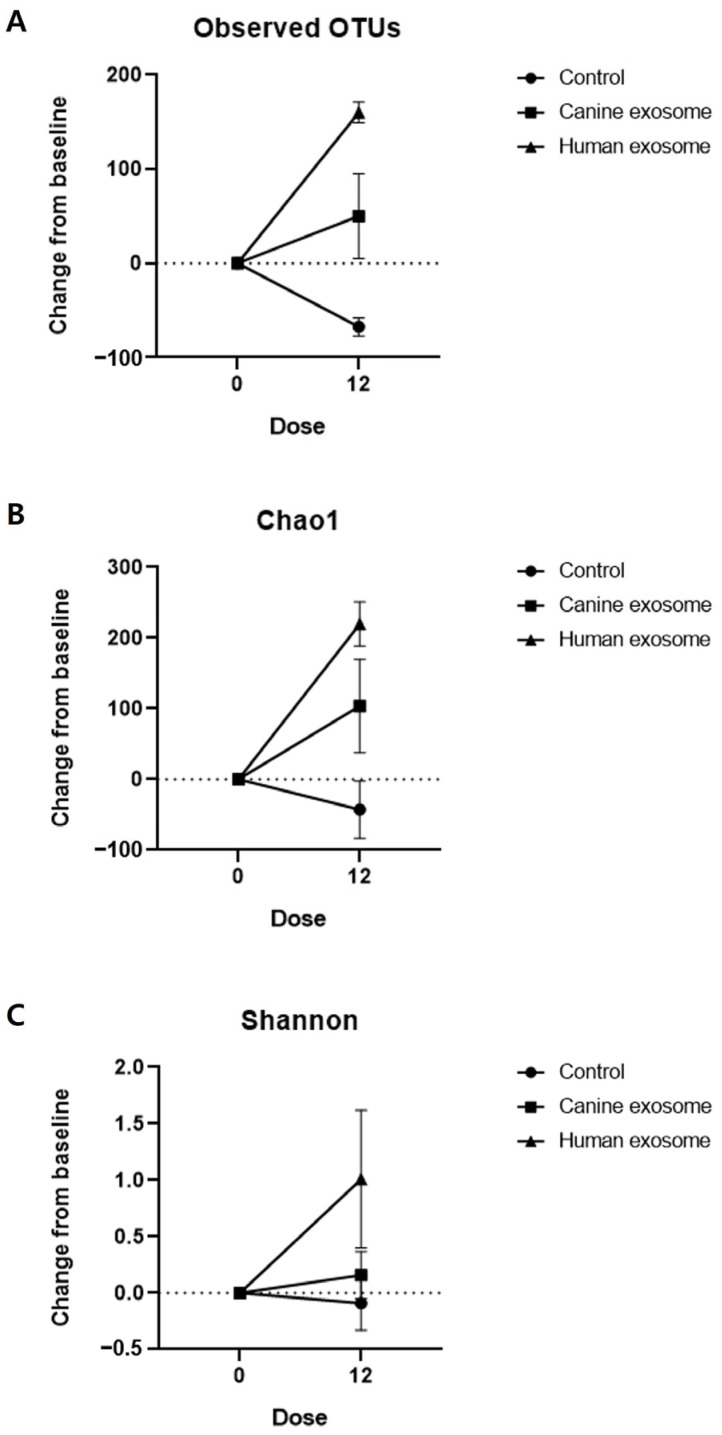
Analysis of alpha diversity of the skin microbiome. Changes in (**A**) observed OTUs, (**B**) Chao1, and (**C**) Shannon indices compared with the baseline values. Dose 0, D0; Dose 12, D28. All values are expressed as the mean ± standard error.

**Figure 7 animals-14-00282-f007:**
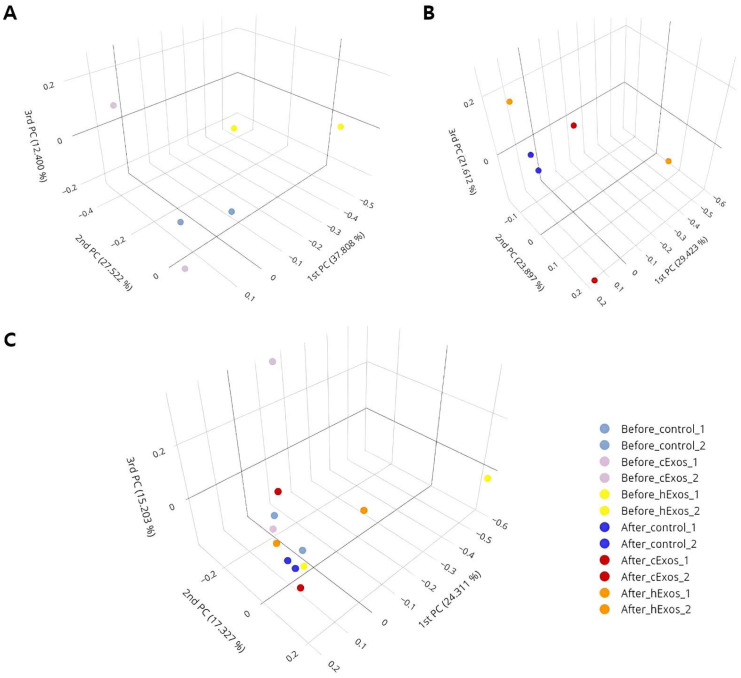
Principal coordinates analysis (PCoA) plot using the generalized UniFrac metric before (**A**) and after treatment (**B**), respectively. PCoA plot of all twelve samples (**C**).

**Table 1 animals-14-00282-t001:** Changes in serum IFN-γ, IL-2, IL-4, IL-10, IL-12, IL-13, IL-31, and TGF-β levels in control, cExos, and hExos groups compared with the baseline levels.

Cytokine (pg/mL)	Control	cExos	hExos
IFN-γ	−21,903 ± 2328	−43,335 ± 5973	−29,327 ± 2188
IL-2	−105,333 ± 1667	−153,000 ± 15,333	−134,833 ± 1500
IL-4	7833 ± 833.4	−27,542 ± 3542	−23,308 ± 891.7
IL-10	884.7 ± 115.4	1060 ± 170.4	1245 ± 91.2
IL-12	−1275 ± 131.7	−1787 ± 107.1	−3271 ± 66.05
IL-13	−575.1 ± 56.25	−2209 ± 908.6	−3859 ± 791.4
IL-31	12,676 ± 369.2	−26,238 ± 4353	−50,364 ± 6135
TGF-β	864.6 ± 5188	13,860 ± 1852	15,424 ± 1401

IFN, interferon; IL, interleukin; TGF, transforming growth factor; cExos, canine exosomes; hExos, human exosomes.

**Table 2 animals-14-00282-t002:** Summary of the alpha diversity indices of the skin microbiome pre and post-treatment.

Group	Subjects	Treatment	Total Valid Reads	Number of Normalized Reads	Observed OTUs	Chao1	Shannon	Good’s Coverage (%)
Control	1	Before	69,584	61,927	613	637.29	3.64	99.86
		After	39,789	30,956	555	634.82	3.79	99.60
	2	Before	86,295	61,927	579	628.58	4.10	99.86
		After	47,260	30,956	502	545.36	3.77	99.76
cExos	1	Before	61,927	61,927	524	532.84	3.63	99.92
		After	43,676	30,956	529	570.54	4.00	99.74
	2	Before	62,883	61,927	524	535.76	3.95	99.92
		After	38,454	30,956	619	705.51	3.90	99.57
hExos	1	Before	63,194	61,927	320	343.61	1.37	99.92
		After	49,001	30,956	491	594.91	2.99	99.59
	2	Before	65,333	61,927	625	653.88	4.06	99.88
		After	30,956	30,956	774	842.57	4.46	99.54

OTUs, operational taxonomic units; cExos, canine exosomes; hExos, human exosomes.

**Table 3 animals-14-00282-t003:** Changes in observed OTUs and Chao1 and Shannon indices in the control, cExos, and hExos groups compared with the baseline values.

	Control	cExos	hExos
Observed OTUs	−67.5 ± 9.5	50.0 ± 45.0	160.0 ± 11.0
Chao1	−42.85 ± 40.38	103.7 ± 66.03	220.0 ± 31.31
Shannon	−0.09 ± 0.24	0.16 ± 0.21	1.01 ± 0.61

OTUs, operational taxonomic units; cExos, canine exosomes; hExos, human exosomes.

## Data Availability

The data presented in this study are available on request from the corresponding author.

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
