# Peer review of "Efficacy of Allogeneic and Xenogeneic Exosomes for the Treatment of Canine Atopic Dermatitis: A Pilot Study"

_animals, 2024, doi:10.3390/ani14020282_

Round 1

Reviewer 1 Report

Comments and Suggestions for Authors

1. Although we know that the cost of dogs as experimental animals is relatively high, there are only 6 dogs used in the experiment in this study, and the sample size of the experiment is too small to make the results convincing.

2. How did the authors determine the dosage of exosomes? Is there any basis for that?

3. Since there were only 2 dogs in each group, the statistical analysis could not meet the requirements of statistical analysis, so there were serious problems in the statistical analysis of this study.

Author Response

Dear Reviewer,

We would like to express our sincere gratitude for your thoughtful and constructive review of our manuscript. Your insightful comments and suggestions have significantly contributed to the improvement of our work. We appreciate the time and effort you dedicated to providing detailed feedback, which has undoubtedly strengthened the quality of our research. Your expertise and valuable input are instrumental in shaping the final version of the manuscript. Thank you for your commitment to the peer-review process; your contributions have been invaluable.

  1. Although we know that the cost of dogs as experimental animals is relatively high, there are only 6 dogs used in the experiment in this study, and the sample size of the experiment is too small to make the results convincing.

Response: Thank you for your valuable feedback. We acknowledge that the sample size in our pilot study, involving six dogs, is relatively small. However, the primary objective of this study was to explore the effectiveness of allogeneic and xenogeneic exosomes in CAD.

Despite the limited sample size, our preliminary findings indicate promising positive outcomes. These initial results provide a foundation for considering larger-scale studies in the future. We recognize the importance of robust statistical power for conclusive results, and we are committed to addressing this limitation in subsequent investigations.

We appreciate your suggestion for larger studies, and we agree that they are warranted to further validate and extend the implications of our initial findings. The positive trends observed in this pilot study underscore the potential significance of allogeneic and xenogeneic exosomes in CAD.

Thank you once again for your insightful comments, and we will incorporate these considerations in our future work.

  1. How did the authors determine the dosage of exosomes? Is there any basis for that?

Response: We appreciate your comments. The exosome dosage in this study was determined based on the effective dose and half-life of exosomes identified in our previous studies, which involved a different disease model, along with an undisclosed biodistribution evaluation. We have added the sentences as follows in materials and method part.

“The exosome dosage in this study was determined based on the effective dose and half-life of exosomes identified in our previous studies [32,34], which involved a different disease model.” (Page 3)

  1. Since there were only 2 dogs in each group, the statistical analysis could not meet the requirements of statistical analysis, so there were serious problems in the statistical analysis of this study.

Response: We appreciate your comment. As stated in the discussion section, the limited sample size of only 2 dogs in each group precluded the ability to establish statistical significance in our analysis. Despite this limitation, the study yielded meaningful results. Therefore, we consider this research as a preliminary study, acknowledging its value. This preliminary study is envisioned to serve as a foundation for future investigations, particularly large-scale studies that will allow for robust statistical validation of the observed trends in allogeneic and xenogeneic exosome efficacy in CAD.

Reviewer 2 Report

Comments and Suggestions for Authors

Dear authors,

Dear Editor.

Thank you for the opportunity to review this manuscript.

Please find my considerations below:

The article is of exceptional quality, showcasing results that can be aptly categorized as a pilot study. As a dermatology specialist, I intend to channel my comments exclusively toward this field, given that my expertise primarily resides within this domain rather than in cellular biology.

The results highlighting a substantial reduction in CADESI and pVas TWEL are particularly noteworthy. Nevertheless, a certain redundancy in presentation becomes evident, as the authors have chosen to replicate these findings in both tables and graphs. I would strongly recommend considering a more streamlined approach, opting for just one of these modes of presentation to heighten clarity and optimize the efficiency in conveying the results. Such a refined strategy promises not only to simplify the interpretation of data but also to ensure a more seamless and concentrated presentation.

Regards

Author Response

Dear Reviewer,

We would like to express our sincere gratitude for your thoughtful and constructive review of our manuscript. Your insightful comments and suggestions have significantly contributed to the improvement of our work. We appreciate the time and effort you dedicated to providing detailed feedback, which has undoubtedly strengthened the quality of our research. Your expertise and valuable input are instrumental in shaping the final version of the manuscript. Thank you for your commitment to the peer-review process; your contributions have been invaluable.

  1. The article is of exceptional quality, showcasing results that can be aptly categorized as a pilot study. As a dermatology specialist, I intend to channel my comments exclusively toward this field, given that my expertise primarily resides within this domain rather than in cellular biology.

The results highlighting a substantial reduction in CADESI and pVas TWEL are particularly noteworthy. Nevertheless, a certain redundancy in presentation becomes evident, as the authors have chosen to replicate these findings in both tables and graphs. I would strongly recommend considering a more streamlined approach, opting for just one of these modes of presentation to heighten clarity and optimize the efficiency in conveying the results. Such a refined strategy promises not only to simplify the interpretation of data but also to ensure a more seamless and concentrated presentation.

Response: Thank you for your valuable feedback. We appreciate your insightful comments and have carefully considered your suggestion. Following your recommendation, we have revised the presentation in Figures 3 and 4. Specifically, we have removed the table, opting to include only the graph to eliminate redundancy. We believe that this modification enhances the clarity of our results and streamlines the presentation for a more efficient interpretation of the data. Your guidance has been invaluable in refining our approach, and we hope these changes align with your expectations.

Reviewer 3 Report

Comments and Suggestions for Authors

This is a review of the manuscript entitled Efficacy of allogeneic and xenogeneic exosomes for the treatment of canine atopic dermatitis: a pilot study. This is a very interesting study evaluating a new and innovative treatment for atopic dogs. Canine atopic dermatitis is one of the major cutaneous diseases affecting dogs, for which there is not a ‘one size fits all’ treatment. The treatment for this disease is still difficult in many cases, so having such research is important. This study does provide substantial new scientific knowledge and can be the first of larger studies.

Introduction:

The concepts are well summarized.

Line 51: Can you please clarify what you mean by ‘common eruptions’?

Line 58: I agree with the authors that most therapies are for controlling the symptoms of atopy and not the disease itself. However, allergen specific immunotherapy (ASIT) may not only control the disease but also possibly cure the disease. I suggest adding a sentence about ASIT, which may result in a better outcome than just controlling the symptoms of the disease.

Materials and Methods:

Lines 93-94: When describing the dogs, you are referring to a previously published study, which is as expected. However, to help the readers to have a better understanding of your subjects, without necessarily having to read the previous study, I would highly suggest adding some information regarding the dogs of this study. E.g., are they usually maintained on antipruritic treatments? ; did you challenge them (if so, with mites? and how (through which route)?) prior to starting this study? ; …

Lines 151-154: The authors have used the CADESI to score the lesions. However, they used this scoring system for only one site. CADESI is a global assessment of the lesions found on the whole dog, not only just one area. I would recommend modifying this part and mentioning that the scale used to evaluate their score was inspired by CADESI, but I think it’s a misuse of this scale as it’s mentioned here.  

Results:

Figure 2: If it’s possible to improve the quality of photos, it would be appreciated.

Skin microbiome analysis: The authors detailed partially the alpha-diversity (richness and diversity index of each sample). Although the alpha-diversity gives some details on the local microbiome, it does not allow comparing between various populations. The beta-diversity must also be calculated and presented (including ideally PCoA plots), to give a complete portrait of the microbiome and its content.

 Discussion:

Lines 262 and 275: typo… ‘antipruritic’ 

Lines 270 – 273: It’s mentioned that pro-inflammatory cytokines and anti-inflammatory cytokines decreased and increased, respectively, in the treated groups. However, in the control group, we can see a similar trend. I suggest that the authors explain or comment this fact, and perhaps give more details about limitations when evaluating circulating cytokines.

Lines 279-280: The complete evaluation of the skin microbiome involves more than what is presented in this study. I think the authors overinterpret their data when they conclude on an improvement of skin microbiome based just on the evaluation of the alpha-diversity on one cutaneous site only.  

 Overall, the discussion is good, and the limitations are well described.

Conclusions:

The conclusions are aligned with the design of the study.

Comments on the Quality of English Language

No comments.

Author Response

Dear Reviewer,

We would like to express our sincere gratitude for your thoughtful and constructive review of our manuscript. Your insightful comments and suggestions have significantly contributed to the improvement of our work. We appreciate the time and effort you dedicated to providing detailed feedback, which has undoubtedly strengthened the quality of our research. Your expertise and valuable input are instrumental in shaping the final version of the manuscript. Thank you for your commitment to the peer-review process; your contributions have been invaluable.

  1. Line 51: Can you please clarify what you mean by ‘common eruptions’?

Response: Thank you for your valuable feedback. We have revised the relevant sentences to enhance clarity. In the revised version (page 2), we now specify, "Primary skin lesions usually consist of erythematous macular or papular eruptions." We hope this provides the necessary clarification regarding the term 'common eruptions.' If you have any further concerns or suggestions, please feel free to let us know.

  1. Line 58: I agree with the authors that most therapies are for controlling the symptoms of atopy and not the disease itself. However, allergen specific immunotherapy (ASIT) may not only control the disease but also possibly cure the disease. I suggest adding a sentence about ASIT, which may result in a better outcome than just controlling the symptoms of the disease.

Response: We appreciate your insightful suggestion. The role of ASIT in the treatment of CAD has already described in lines 272–275.

  1. Lines 93-94: When describing the dogs, you are referring to a previously published study, which is as expected. However, to help the readers to have a better understanding of your subjects, without necessarily having to read the previous study, I would highly suggest adding some information regarding the dogs of this study. E.g., are they usually maintained on antipruritic treatments? ; did you challenge them (if so, with mites? and how (through which route)?) prior to starting this study? ; …

Response: We appreciate your comments. In accordance with your suggestions, we have added the sentences as follows.

“Briefly, the dogs were epicutaneously sensitized and challenged with house dust mite, Dermatophagoides farinae, for 12 weeks and 4 weeks, respectively. Immediately after the completion of the mite challenge, six CAD model dogs were included.” (Page 2)

  1. Lines 151-154: The authors have used the CADESI to score the lesions. However, they used this scoring system for only one site. CADESI is a global assessment of the lesions found on the whole dog, not only just one area. I would recommend modifying this part and mentioning that the scale used to evaluate their score was inspired by CADESI, but I think it’s a misuse of this scale as it’s mentioned here.

Response: We appreciate your comments. In accordance with your suggestions, we have corrected the sentences as follows.

Skin lesion score and transepidermal water loss decreased in cExos and hExos groups compared with those in the control group. (Page 1)

On day 0 (D0), skin lesion scoring was performed using a scale that was a modified form of canine atopic dermatitis extent and severity index (CADESI-04), and transepidermal water loss (TEWL) was measured. (Page 3)

Lesion scoring was performed after four and eight doses of the treatments and at the end of the experiment using the modified CADESI-04 scale [35]. (Page 4)

During the study, lesion scoring was performed after 4 and 8 doses of treatment (before the injection on D10 and D19) using the modified CADESI-04 scale.” (Page 4)

The severity of skin lesions was scored on days 0, 10, 19, and 28 using a modified CADESI-04 scale. (Page 4)

Additionally, cExos and hExos treatments caused a gradual decrease in the modified CADESI-04 score. Specifically, the change in modified CADESI-04 score …” (Page 5)

“Changes in lesion scores at different time points compared with the baseline values.” (Page 6)

“Treatment with cExos and hExos ameliorated skin lesions in dogs with CAD compared with that in the control group, as evidenced by a decrease in modified CADESI-04 scores in the cExos and hExos groups.” (Page 10)

  1. Figure 2: If it’s possible to improve the quality of photos, it would be appreciated.

Response: We appreciate your insightful comments. Taking your suggestion into consideration, we have increased the DPI of Figure 2 to enhance the quality of the image. The revised figure with improved resolution is now included in the manuscript. We hope this meets your expectations, and we remain open to any further suggestions or feedback you may have.

  1. Skin microbiome analysis: The authors detailed partially the alpha-diversity (richness and diversity index of each sample). Although the alpha-diversity gives some details on the local microbiome, it does not allow comparing between various populations. The beta-diversity must also be calculated and presented (including ideally PCoA plots), to give a complete portrait of the microbiome and its content.

Response: We appreciate your comments. In accordance with your suggestions, we have added the sentence and figure as follows.

“Also, principal coordinate analysis (PCoA) plots from generalized UniFrac metric were constructed (Figure 7).” (Page 8)

  1. Lines 262 and 275: typo… ‘antipruritic’

Response: We appreciate your comments. In accordance with your suggestions, we have corrected the sentences as follows.

“Lokivetmab showed more pronounced antipruritic efficacy than cyclosporine but had a lower anti-inflammatory effect” (Page 10)

“Given that IL-31 is a potent pruritogenic cytokine, the decrease in IL-31 levels emphasizes the antipruritic effect of exosomes” (Page 11)

  1. Lines 270 – 273: It’s mentioned that pro-inflammatory cytokines and anti-inflammatory cytokines decreased and increased, respectively, in the treated groups. However, in the control group, we can see a similar trend. I suggest that the authors explain or comment this fact, and perhaps give more details about limitations when evaluating circulating cytokines.

Response: We appreciate your comments. In accordance with your suggestions, we have added the sentences as follows.

“Moreover, exosome treatments decreased the serum levels of inflammatory cytokines, including IFNγ, IL-2, IL-4, IL-12, IL-13, and IL-31, and increased the levels of anti-inflammatory cytokines, including IL-10 and TGFβ. This trend was more pronounced in the exosome treatment groups than in the control group, indicating that exosome therapy has an immunomodulatory effect.” (Page 10)

  1. Lines 279-280: The complete evaluation of the skin microbiome involves more than what is presented in this study. I think the authors overinterpret their data when they conclude on an improvement of skin microbiome based just on the evaluation of the alpha-diversity on one cutaneous site only.

Response: Thank you for your insightful comments. In our study, we employed a focal CAD experimental model, which led us to focus specifically on the skin microbiome in the right groin area. We acknowledge the limitation of not including histopathology and immunohistochemistry in our analysis, and this is discussed in the limitations section of the manuscript.

Reviewer 4 Report

Comments and Suggestions for Authors

Thank you for the strong paper regarding exosomal therapies. I wonder why you did not describe the production of the exosomal (human and canine) production within the study. I understand it was previously published but having this overview of the therapy process would be beneficial. 

Why was the sample size so low? 

Author Response

Dear Reviewer,

We would like to express our sincere gratitude for your thoughtful and constructive review of our manuscript. Your insightful comments and suggestions have significantly contributed to the improvement of our work. We appreciate the time and effort you dedicated to providing detailed feedback, which has undoubtedly strengthened the quality of our research. Your expertise and valuable input are instrumental in shaping the final version of the manuscript. Thank you for your commitment to the peer-review process; your contributions have been invaluable.

  1. Thank you for the strong paper regarding exosomal therapies. I wonder why you did not describe the production of the exosomal (human and canine) production within the study. I understand it was previously published but having this overview of the therapy process would be beneficial.

Response: We appreciate your comments. In accordance with your suggestions, we have added the sentences as follows.

“Allogeneic exosomes (cExos) used in this study were isolated from canine adipose tissue-derived MSCs. The xenogeneic exosomes (hExos) were Expi293F cell-derived exosomes loaded with super-repressor IκB (srIκB). cExos and hExos were obtained following the methods reported in previous studies [31–33]. Briefly, both upstream and downstream processes were used in this study. In the upstream process, the cells were cultivated for 4 days using the WAVE method under blue-light exposure. Afterward, the conditioned media was subjected to ultrafiltration and purification to obtain the purified exosomes in the downstream process. (Page 3)

  1. Why was the sample size so low?

Response: Thank you for your insightful comment. The limited sample size in this study, attributed to budget constraints, involved six dogs. Despite this constraint, our pilot study yielded meaningful results, confirming the therapeutic potential of allogeneic and xenogeneic exosomes for CAD through various analyses. We view this preliminary study as valuable, given the obtained results within the confines of the research budget. This work thus lays the foundation for future investigations, with a focus on larger-scale studies to robustly validate observed trends in exosome efficacy for CAD.

Round 2

Reviewer 1 Report

Comments and Suggestions for Authors

There are serious design problems with this study that are not addressed in the author's revised draft. Due to the small sample size of the study, the results and conclusions are unreliable. Therefore, I recommend rejection.

Author Response

Thank you for your valuable feedback. We acknowledge that the sample size in our pilot study, involving six dogs, is relatively small. However, the primary objective of this study was to explore the effectiveness of allogeneic and xenogeneic exosomes in CAD.

Despite the limited sample size, our preliminary findings indicate promising positive outcomes. These initial results provide a foundation for considering larger-scale studies in the future. We recognize the importance of robust statistical power for conclusive results, and we are committed to addressing this limitation in subsequent investigations.

We appreciate your suggestion for larger studies, and we agree that they are warranted to further validate and extend the implications of our initial findings. The positive trends observed in this pilot study underscore the potential significance of allogeneic and xenogeneic exosomes in CAD.

Thank you once again for your insightful comments, and we will incorporate these considerations in our future work.